# Dynamic Parameters of Hypothermic Machine Perfusion—An Image of Initial Graft Function in Adult Kidney Transplantation?

**DOI:** 10.3390/jcm11195698

**Published:** 2022-09-27

**Authors:** Sebastian Weberskirch, Shadi Katou, Stefan Reuter, Felicia Kneifel, Mehmet Haluk Morgul, Felix Becker, Philipp Houben, Andreas Pascher, Thomas Vogel, Sonia Radunz

**Affiliations:** 1Department of General, Visceral and Transplant Surgery, University Hospital Muenster, 48149 Muenster, Germany; 2Department of Internal Medicine, Nephrology and Rheumatology, University Hospital Muenster, 48149 Muenster, Germany

**Keywords:** arterial flow, cold ischemic time, delayed graft function, hypothermic machine perfusion, organ resistance, static cold storage

## Abstract

Kidney allografts are subjected to ischemia reperfusion injury during the process of transplantation. Hypothermic machine perfusion (HMP) of deceased donor kidneys from organ procurement until transplantation is associated with a superior outcome when compared to static cold storage (SCS). Nevertheless, cold ischemia time (CIT) remains an independent risk factor for delayed graft function (DGF) in HMP-preserved kidney allografts as well. We performed a retrospective single-center study including all adult recipients who underwent deceased donor kidney-only transplantation at our center between January 2019 and December 2020. Beside the clinicopathological donor and recipient data, flow and resistance data during HMP were assessed. Short- and long-term kidney allograft outcome after end-ischemic HMP and SCS were analyzed and compared. Organ preservation consisted of either SCS (*n* = 88) or HMP (*n* = 45). There were no differences in recipient demographics and donor details between groups. CIT was significantly longer in the HMP group (16.5 [8.5–28.5] vs. 11.3 [5.4–24.1], *p* < 0.0001). The incidence of DGF as well as serum creatinine at discharge and at 1 year post transplant were comparable between groups. Duration of SCS prior to HMP was comparable among grafts with and without DGF. Flow rate and organ resistance at the start of HMP were significantly worse in DGF-kidney grafts (arterial flow 22.50 [18.00–48.00] vs. 51.83 [25.50–92.67] ml/min, *p* = 0.0256; organ resistance 123.33 [57.67–165.50] vs. 51.33 [28.17–111.50] mmHg/mL/min, *p* = 0.0050). Recipients with DGF had significantly worse creatinine levels at discharge (2.54 [1.08–7.64] vs. 1.67 [0.90–6.56], *p* < 0.0001) and at 1 year post transplant (1.80 [1.09–7.95] vs. 1.59 [0.87–7.40], *p* = 0.0105). In conclusion, baseline HMP parameters could be applied as a predictive tool for initial graft function, which in turn determines long-term outcome.

## 1. Introduction

Ischemia reperfusion injury (IRI) is inevitable in kidney transplantation and one of the most important mechanisms for impaired early graft function [1]. Ischemia reperfusion injury consists of complex pathophysiological mechanisms, involving activation of cell death programs, endothelial dysfunction, transcriptional reprogramming and activation of the innate and adaptive immune system [2,3]. Anaerobic metabolism during cold ischemia results in the accumulation of toxic substances, lysosomal instability and cellular edema through inhibition of Na^+^/K^+^ ATPase pumps [4]. Increasing oxygen levels and normalizing pH during reperfusion is then harmful for the ischemic cells [1,3]. 

Cold ischemia time (CIT) is a known major risk factor for IRI [3]. Particularly prolonged CIT is directly associated with delayed graft function (DGF), predisposing to inferior one-year graft function and decreased graft and patient survival in kidney transplantation [5,6,7,8,9]. The documented incidence of DGF varies widely (24–70%), depending on the definition used [9].

Especially in extended criteria donor kidney allografts, optimal organ preservation from procurement to transplantation is of utmost importance for short- and long-term graft function [10]. Hypothermic machine perfusion (HMP) of deceased donor kidneys initiated at the donor hospital is associated with superior outcome when compared to static cold storage (SCS) [11]. The positive effect of HMP may be attributed to vasodilatation and a lower vascular resistance of the kidney allograft after HMP, resulting in improved oxygen and nutrient provision after reperfusion in the recipient [12]. However, the benefits of short CIT still outweigh the benefits of HMP, and CIT remains an independent risk factor for DGF even in kidney allografts undergoing preservation-HMP [13]. If HMP is only initiated in the recipient hospital, SCS prior to HMP sometimes lasts several hours and may have an even greater impact on short- and long-term graft function.

Kidney allograft assessment regarding the prediction of DGF is commonly based on donor profiles and histological evaluation of zero-time biopsies; nevertheless, the final decision to accept an organ for transplantation remains a subjective one. Objective and reliable techniques in kidney graft viability assessment prior to transplantation are not established yet. Dynamic kidney preservation offers the opportunity to assess potential markers that may portray information on organ viability and quality [14,15,16].

The objective of the present study was to evaluate the effect of organ preservation on initial graft function and to assess the potential predictive value of HMP parameters for DGF in deceased-donor kidney transplantation.

## 2. Materials and Methods

All adult recipients who underwent deceased-donor kidney transplantation at our transplant center between January 2019 and December 2020 were included in the study. The study was conducted in accordance with the declaration of Helsinki and after obtaining the approval of the local ethics committee (ID 2021-223-f-S). The prerequisite for written informed consent was waived since the study was a retrospective chart analysis.

Following organ procurement, kidney allografts were preserved by SCS using histidine–tryptophan–ketoglutarate solution (Custodiol^®^ HTK Solution, Dr. Franz Köhler Chemie GmbH, Bensheim, Germany) or University of Wisconsin solution (Belzer UW^®^ Cold Storage Solution, Bridge to Life (Europe) Ltd., London, United Kingdom) during transportation from the procurement hospital to the recipient hospital. Further organ preservation consisted of either SCS or HMP, depending on factors resulting in a prolonged CIT, e.g., recipients’ need for dialysis and/or plasmapheresis, operating room capacity.

After back table preparation, kidneys in the HMP group were connected to the Lifeport^®^ Kidney Transporter (Organ Recovery Systems, Chicago, IL, USA) and perfused at 2–4 °C using one liter of Kidney Preservation Solution-1 (Organ Recovery Systems Chicago, IL, USA). Monitoring of the allografts during HMP consisted of automatic recording of temperature, flow, vascular resistance and infusion pressure. The Lifeport^®^ software calculates flow and resistance every 10 s; based on previous publications reporting on HMP parameters, we chose to analyze these parameters at the beginning of perfusion, after 1 h, at mid-point and at the end of perfusion. Mean values of the recordings over one minute were used for analysis [17].

Kidney allografts were transplanted in the iliac fossa with end-to-side vascular anastomoses to the external iliac vessels. Ureteroneocystostomy was performed according to the modified Lich–Gregoir technique and with insertion of a double-J stent. According to local protocol, patients received Thymoglobulin^®^ 1.5 mg/kg for immunosuppressive induction on the day of transplantation. Depending on cellular immune status, further dosages of Thymoglobulin^®^ were administered on postoperative days (POD) 1–3. Maintenance immunosuppression consisted of tacrolimus (trough level 6–8 ng/mL until month 3, then 5–7 ng/mL), mycophenolate-mofetil and steroids.

The following recipient characteristics were assessed: age, sex, body mass index (BMI) and time on dialysis. The following donor variables were collected: age, sex, BMI, history of hypertension and/or diabetes mellitus, hepatitis C virus (HCV) infection status, cause of brain death, serum creatinine (sCr) levels at procurement and kidney donor risk index (KDRI) [18]. Extended criteria donors (ECD) were defined using the United Network for Organ Sharing (UNOS) definition: (a) donors >60 years, or (b) donors 50–59 years with at least 2 of the following: sCr >1.5 mg/dL, history of hypertension or cardiovascular death [19]. The following procurement data and surgical details were evaluated: method of organ preservation, HMP duration, CIT and warm ischemia time (WIT). The following outcome variables were recorded: sCr levels on POD 1, 3, 7, on the day of discharge and at 1 year post transplant, presence of DGF, patient and graft survival. DGF was defined as the need for at least one hemodialysis session during the first week of posttransplant [9].

All data were tested for normality using the D’Agostino and Pearson omnibus normality test. Categorical variables were presented as percentages and continuous variables as median [range], unless stated otherwise. Differences between categorical variables were tested using Fisher’s exact test or chi-square test as appropriate. Differences in continuous variables were tested using Student’s t-test or Mann–Whitney test as appropriate. The reference point for all calculations of survival was the day of kidney transplantation. Overall graft survival was determined until patient death, return to dialysis or the end of the study period. A *p* value ≤ 0.05 (two-tailed) was considered to be significant. Data collection and statistical analysis were performed using Microsoft Excel 2010 (Microsoft Corporation, Redmond, WA, USA) and GraphPad Prism 9 for macOS version 9.3.1 (GraphPad Software, San Diego, CA, USA). Logistic regression analysis was performed using IBM SPSS Statistics (version 27.0 for Windows, SPSS, Inc., Chicago, IL, USA).

## 3. Results

### 3.1. Demographic Data

Between January 2019 and December 2020, 200 patients underwent kidney transplantation at our transplant center. Children under the age of 18 (*n* = 13), recipients of combined-organ transplants (*n* = 15) and recipients of living donation (*n* = 39) were excluded from this study. 

Recipients’ demographics are given in Table 1. Organ preservation was performed by SCS in the majority of the study population (*n* = 88, 66.2%); HMP was applied in 45 (33.8%) donor kidneys. Recipient and donor characteristics did not differ significantly with regards to preservation method, besides dialysis vintage being significantly longer in the HMP group (*p* = 0.03). CIT was significantly longer in the HMP group with the majority lasting more than 15 hrs; in the majority of the SCS group, CIT was less than 15 hrs (Table 2). SCS prior to HMP lasted 6.3 [0.2–19.1] hrs. WIT was however significantly shorter in the HMP group (*p* = 0.0055)

Incidence of DGF was high in our entire study population (*n* = 57, 42.9%), but comparable between groups (SCS group 43.2% vs. HMP group 42.2%, *p* > 0.9999). Twelve patients received only one dialysis treatment within the first seven postoperative days, nine of these patients on POD 0 or 1 (SCS group *n* = 4, HMP group *n* = 5).

### 3.2. Outcome Data

Initial creatinine and creatinine at 1 year post transplant were comparable between both groups (Figure 1). In the SCS group, transplant nephrectomy had to be performed in three patients on POD 5 (graft thrombosis), 38 (refractory rejection) and 258 (bleeding) after kidney transplant; in the HMP group, transplant nephrectomy had to be performed in one patient on POD 6 due to graft thrombosis (*p* = 0.3132).

Patient survival was 96.2% in our study population and study period. In the SCS group, two patients died on POD 21 and POD 32 due to multiorgan failure, one patient died of unknown causes on POD 355, and one patient died due to SARS-CoV-2 pneumonia on POD 592 with a functioning graft. In the HMP group, one patient died due to pneumonia on POD 16 with a functioning graft.

Duration of SCS prior to HMP was comparable among grafts with and without DGF (6.2 [3.9–19.1] vs. 6.6 [0.4–13.0] h, *p* = 0.7717). Cardiovascular risk factors of donors, such as hypertension (24.6% vs. 38.2%, *p* = 0.1337) and diabetes (7.1% vs. 15.8%, *p* = 0.1050), were comparable between groups. Male donor sex was significantly more frequent in recipients with DGF, and there was a trend for a higher recipient BMI in recipients receiving DGF grafts (Table 3). Multivariable analysis only detected donor male sex as independently associated with the occurrence of DGF (Table 4). 

Patients with DGF demonstrated significantly higher creatinine levels at discharge (2.54 [1.08–7.64] vs. 1.67 [0.90–6.56], *p* < 0.0001) and at 1 year post transplant (1.80 [1.09–7.95] vs. 1.59 [0.87–7.40], *p* = 0.0105). Graft loss was comparable among recipients with and without DGF (5.3% vs. 1.3%, *p* = 0.3132). Subgroup analysis of DGF grafts vs. immediately functioning grafts in the SCS group and in the HMP group, respectively, did not yield differences in donor and recipient characteristics in these subgroups (Table 5).

### 3.3. HMP Data

Kidney allografts with immediate function exhibited significantly higher arterial flow and significantly lower organ resistance at HMP start. Both grafts with and without DGF demonstrated a significant increase in arterial flow and a significant decrease in organ resistance already after one hour of HMP. For the further course, arterial flow and organ resistance were comparable between groups (Figure 2 and Figure 3). The improvement of perfusion parameters was independent of the duration of HMP.

## 4. Discussion

SCS is still widely in use for organ preservation due to its simple and cost-effective applicability despite evidence for improved graft function of HMP-preserved kidney allografts [20]. Frequently, HMP is only initiated in the recipient hospital. However, the typical deceased kidney donor today is of advanced age and has concomitant vascular diseases; these factors may have a negative effect on allograft quality and result in an impaired initial graft function, especially after prolonged SCS. Kidney allografts with an impaired initial graft function carry an increased risk of acute rejection, and inferior long-term graft survival [8].

Furthermore, the extent of the beneficial effects of HMP may depend on the applied definition of DGF, and the thresholds for the use of dialysis may vary between centers. Mallon et al. suggested that the most widely used and most easily calculated definition, i.e., the need for dialysis within the first seven postoperative days, should be universally adopted as the definition of DGF clinically, and as a study endpoint [9]. The effect of HMP on the incidence of acute rejection, patient survival, hospital stay, long-term graft function and duration of DGF could not yet be determined [20]. Furthermore, knowledge about the exact mechanism of how HMP improves kidney outcome, is scarce. 

In our small study cohort, the acknowledged advantage of HMP for the reduction of DGF [10] did not become evident. Neither HMP nor short CIT resulted in a lower incidence of DGF. Nevertheless, HMP may have compensated for prolonged CITs, as the significantly prolonged CIT in the HMP group did not negatively affect DGF occurrence.

HMP initiated directly after procurement was proven to result in a significantly improved 1 year graft survival in kidney transplantation compared to SCS [11]. Against general assumptions, HMP from procurement to transplantation actually yielded the largest benefit in kidneys which were preserved for a relatively short period, i.e., less than 10 h of CIT. Kox et al. demonstrated that the detrimental effect of CIT during HMP has a similar magnitude as during SCS; regardless of the preservation method, every additional hour of CIT increased the odds for developing DGF by 8% [13]. In registry data of kidney allografts mainly preserved by end-ischemic HMP, no significant benefit of HMP after a short CIT was demonstrated [21]. Hence, the effect of CIT in HMP-preserved kidneys is still a topic of debate.

A Cochrane review identified that current studies fail to provide evidence of a benefit of HMP in terms of graft non-function, and research investigating the use of perfusion parameters to perform viability assessment may be useful in preventing graft non-function [20]. Adani et al. demonstrated that even a delayed HMP may recover allograft hemodynamic function, maintain some metabolic activity and stabilize the accumulated ischemic damage due to initial SCS [17]. After a minimum of 6 h HMP, a flow >80 mL/min and an organ resistance of <0.3 mmHg/mL/min are considered thresholds for utilization [22]. 

In our study, improvement of arterial flow and organ resistance was present already after one hour of HMP in kidney allografts with and without DGF alike; only baseline values demonstrated an increased resistance and reduced flow in DGF kidneys. This is in accordance with results published by Patel et al. achieving average flow rates in the first two hours after perfusion [23]. Despite improvement in perfusion parameters, post-transplant graft function is more likely related to baseline arterial flow and organ resistance, although one does expect that perfusion improvement would facilitate improved oxygen and nutrient provision after reperfusion in the recipient.

Knowledge about the exact mechanism of kidney transplant outcome improvement after pulsatile HMP is still limited. Pulsatile flow is known to prevent vasospasm and to promote vasodilation upon reperfusion; improved microperfusion at the time of reperfusion increases the chances of immediate graft function [24]. Nevertheless, cellular metabolism is not entirely ceased during HMP, and oxygen deficiency results in mitochondrial dysfunction and production of ROS [25]. Besides, HMP does not have a re-conditioning effect, and ECD kidney allografts especially may still suffer from detrimental effects of prolonged CIT. The baseline HMP parameters could be a valuable additive tool to evaluate graft quality and predict DGF. Nevertheless, we explicitly refrain from endorsing cut-off values for organ acceptance. In times of severe organ shortage, rejection of organs for transplantation based on not yet established perfusion parameters might lead to a severe increase in waiting times and consecutively an increase in recipient morbidity.

The present study has some limitations inherent in any retrospective study design with a limited number of patients. Due to the typical application of HMP only at the recipient center, a subgroup analysis of HMP during transport versus HMP at the recipient center alone was not completed. We applied the most widely used definition of DGF, resulting in a high incidence in the entire study population. Protective factors independently associated with an improved outcome were not detected in this small sample size.

In conclusion, end-ischemic HMP outweighs the effects of significantly prolonged CIT. Furthermore, baseline HMP parameters could render a predictive tool for initial graft function. Differences in short- and long-term outcomes of kidney transplantation in the setting of prolonged CIT might be more evident in preservation-HMP, and studies analyzing HMP during transport versus at the recipient center alone are still warranted.

## Figures and Tables

**Figure 1 jcm-11-05698-f001:**
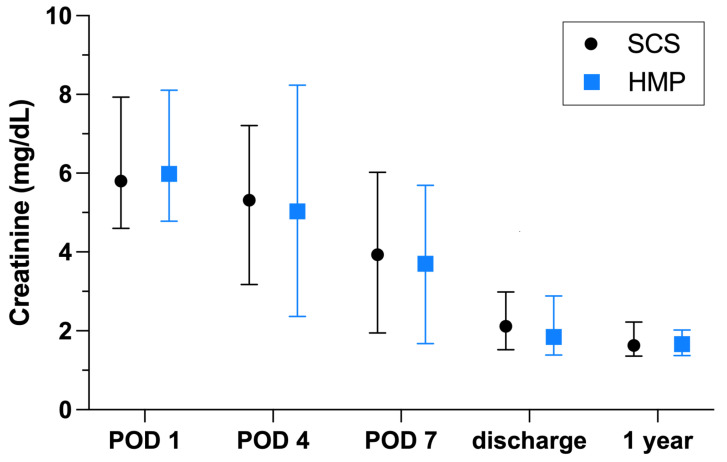
Creatinine levels according to preservation method. Data are presented as median (interquartile range). Static cold storage, SCS; hypothermic machine perfusion, HMP.

**Figure 2 jcm-11-05698-f002:**
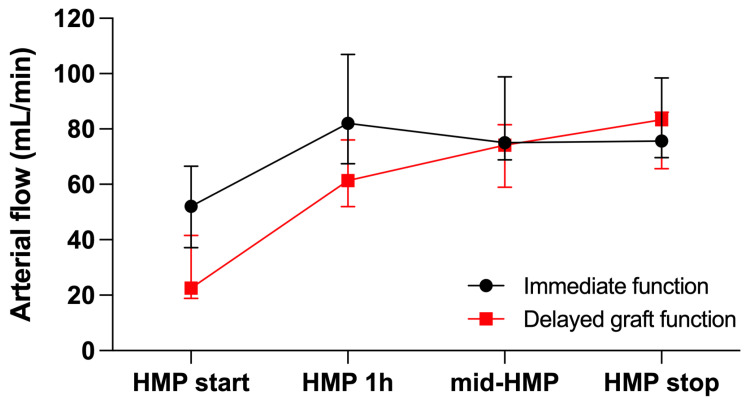
Arterial flow during hypothermic machine perfusion (HMP) according to graft function. Data are presented as median (interquartile range).

**Figure 3 jcm-11-05698-f003:**
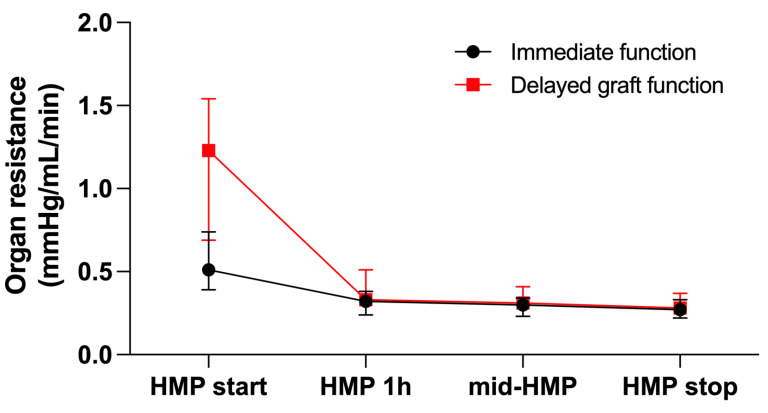
Organ resistance during hypothermic machine perfusion (HMP) according to graft function. Data are presented as median (interquartile range).

**Table 1 jcm-11-05698-t001:** Patients’ demographics, donor and organ allocation details.

	SCS (n = 88)	HMP (n = 45)	*p*
Recipient age (years)	60 [22–80]	60 [26–81]	0.9669
Recipient male sex (%)	58.0	66.7	0.3541
Recipient BMI (kg/m^2^)	25.9 [18.2–40.0}	26.2 [18.0–38.2]	0.8469
Dialysis vintage (months)	76.7 [11.7–159.9]	84.6 [11.9–186.0]	0.0310
Donor age (years)	58 [0–85]	56 [19–84]	0.8450
Donor male sex (%)	53.4	66.7	0.1937
Donor BMI (kg/m^2^)	25.0 [12.8–43.6]	24.9 [15.2–43.6]	0.9349
KDRI	1.26 [0.57–2.96]	1.25 [0.73–2.55]	0.9783
ECD (%)	50.0	48.9	>0.9999

static cold storage, SCS; hypothermic machine perfusion, HMP; body mass index, BMI; kidney donor risk index, KDRI; extended criteria donor, ECD.

**Table 2 jcm-11-05698-t002:** Surgical details.

	SCS (n = 88)	HMP (n = 45)	*p*
Cold ischemia time (h)	11.3 [5.4–24.1]	16.5 [8.5–28.5]	<0.0001
0–10 h (%)	36.4	4.4	
10–15 h (%)	40.9	20.0	
15–20 h (%)	17.0	46.7	
>20 h (%)	5.7	28.9	
Duration of HMP (h)	n.a.	10.1 [3.4–18.3]	
Warm ischemia time [min]	39 [15–70]	34 [16–65]	0.0055

static cold storage, SCS; hypothermic machine perfusion, HMP.

**Table 3 jcm-11-05698-t003:** Recipient, donor and surgical details with regards to initial graft function.

	DGF (*n* = 57)	Immediate Function (*n* = 76)	*p*
Recipient age (years)	60 [22–81]	60 [26–80]	0.5965
Recipient male sex (%)	66.7	56.6	0.2129
Recipient BMI (kg/m^2^)	26.7 [18.1–40.0}	25.3 [18.0–37.7]	0.0521
Dialysis vintage (months)	80.9 [28.2–150.2]	77.6 [11.7–186.0]	0.2537
Donor age (years)	56 [0–85]	59 [6–85]	0.5145
Donor male sex (%)	68.4	50.0	0.0355
Donor BMI (kg/m^2^)	24.8 [12.8–43.6]	25.2 [12.8–43.6]	0.9395
KDRI	1.21 [0.57–2.46]	1.36 [0.64–2.96]	0.5160
HMP (%)	33.3	34.2	>0.9999
CIT (h)	13.3 [5.4–28.5]	13.3 [5.5–26.3]	0.5064
CIT > 15 h (%)	42.1	39.5	0.8587
WIT [min]	39 [20–70]	35 [15–65]	0.2972

delayed graft function, DGF; body mass index, BMI; kidney donor risk index, KDRI; hypothermic machine perfusion, HMP; cold ischemia time, CIT; warm ischemia time, WIT.

**Table 4 jcm-11-05698-t004:** Results from multivariable logistic regression analysis of risk factors independently associated with delayed graft function.

	Odds Ratio (OR)	95% Confidence Interval	*p*
Recipient age (per 10 years)	1.245	0.880, 1.762	0.216
Recipient BMI	1.072	0.991, 1.160	0.083
Dialysis vintage (per 12 months)	1.107	0.966, 1.268	0.143
Donor male sex	2.223	1.015, 4.868	0.046
KDRI	0.949	0.413, 2.183	0.903
CIT	1.033	0.945, 1.130	0.475
WIT (per 10 min)	0.577	0.214, 1.557	0.277
HMP	1.074	0.754, 1.531	0.692

body mass index, BMI; kidney donor risk index, KDRI; cold ischemia time, CIT; warm ischemia time, WIT; hypothermic machine perfusion, HMP.

**Table 5 jcm-11-05698-t005:** Recipient, donor and surgical details with regards to the preservation method.

	DGF—SCS (*n* = 38)	DGF—HMP (*n* = 19)	*p*
Recipient age (years)	59 [22–76]	60 [44–81]	0.8634
Recipient male sex (%)	60.0	83.3	0.2359
Recipient BMI (kg/m^2^)	26.4 [19.7–40.0}	28.4 [18.1–38.2]	0.6670
Dialysis vintage (days)	80.7 [28.2–145.4]	88.0 [28.2–150.2]	0.1678
Donor age (years)	58 [0–85]	56 [19–84]	0.8174
Donor male sex (%)	65.8	73.7	0.7633
Donor BMI (kg/m^2^)	24.8 [12.8–36.7]	24.9 [15.2–43.6]	0.4370
KDRI	1.23 [0.57–2.21]	1.17 [0.84–2.46]	0.9900
ECD (%)	47.4	42.1	0.7824
CIT (h)	10.9 [5.4–21.2]	16.3 [12.0–28.5]	<0.0001
CIT > 15 h (%)	26.3	73.7	0.0014
WIT [min]	40 [22–70]	33 [20–60]	0.1876

static cold storage, SCS; hypothermic machine perfusion, HMP; delayed graft function, DGF; body mass index, BMI; kidney donor risk index, KDRI; extended criteria donor, ECD; cold ischemia time, CIT; warm ischemia time, WIT.

## Data Availability

The data that support the findings of this study are available from the corresponding author upon reasonable request.

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
