# Peer review of "Dynamic Parameters of Hypothermic Machine Perfusion—An Image of Initial Graft Function in Adult Kidney Transplantation?"

_jcm, 2022, doi:10.3390/jcm11195698_

Round 1

Reviewer 1 Report

In the article titled: Dynamic parameters of hypothermic machine perfusion – an 2 image of initial graft function in adult kidney transplantation, Sebastian Weberskirch et.al described their findings regarding clinical allograft outcomes following static cold storage(SCS) vs hypothermic machine perfusion(HMP). This was a retrospective analysis of recipients of deceased kidney transplants whose organs initially received SCSin the donor hospital but afterward continued HMP or SCS storage of organs in the recipient hospital. The authors reported no significant differences between the groups regarding initial/One-year serum creatinine, graft survival, and patient survival. The authors however noted reduced arterial flow and higher organ resistance in HMP stored organs with DGF compared to those without. This is despite similar pre-SCS cold ischemia time. 

The study investigator made a good attempt to evaluate the benefit of HMP on long-term kidney allograft function. The author acknowledged their limitations including the retrospective nature of the study, small study participants as well as the lack of HMP storage in the donor hospital. 

Overall, the study was well done with a sound scientific background in support of their findings. The study should clarify the following to help the transplant community benefit from their work regarding HMP perfused organs:

1. In view of the average donor age of 56 yrs, are there comorbid factors especially cardiovascular that explain lower arterial flow or higher organ resistance in the donors of patients with DGF?

2. The multivariate analysis showed the overall risk for DGF  with male donors in the overall study population. Are there differences in gender of those with DGF specifically in the HMP stored organs?

3.  The authors should comment on the application of their findings to the overall clinical scenario of a kidney transplant.

a. Should organs with the poor initial arterial flow or higher organ resistance be rejected for transplant

b. What should be the minimum acceptable criteria of this variable that predicts favorable allograft function?

Author Response

We thank the reviewer for the thorough and overall favorable assessment of our work and the important comments made.

  1. Common risk factors for cardiovascular macroangiopathy were comparable among donors of patients with and without DGF (hypertension 24.6% vs. 38.2%, p=0.1337; diabetes 7.1% vs. 15.8%, p=0.1050). We added those factors to our results in the script. 
  2. In HMP preserved kidney allografts alone, male donor sex was comparable among organs developing DGF and those with immediate function (73.7% vs. 61.5%, p=0.5259). Since this was not significant, we did not include it in our results.
  3. The authors believe any organ should be critically evaluated for transplantation; especially in times of severe organ shortage, rejection of organs for transplantation severely increases waiting time and recipient morbidity. In this small study cohort, cut-off values for arterial flow and organ resistance may not be validated. Certainly, transplant surgeons and transplant nephrologists must be aware of perfusion parameters indicative for an impaired organ function, as presented in this study, and should choose possible transplant recipients carefully. Furthermore, transplant professionals should accommodate the post-operative treatment of recipients receiving such organs. We added this point to our discussion.

Reviewer 2 Report

This work would benefit from adding statistical analysis to assess the machine perfusion parameters as risk factors for DGF and graft function at later time. This would provide a better insight whether those parameters could predict DGF and long-term graft function.

Author Response

We thank the reviewer for the overall favorable assessment of our work. In kidney transplantation, delayed graft function (DGF) is a well-established risk factor for impaired long-term graft survival. Likewise, we were able to confirm this common knowledge in our study cohort (significantly increased serum creatinine at discharge and at 1-year post transplant, see page 5, lines 183-184). Thus, perfusion parameters indicative for developing DGF (see page 6, lines 193 ff.) would be indicative for long-term impaired graft function as well.